# G-LLaVA 🦊: Solving Geometric Problem with Multi-Modal Large Language Model

**Jiahui Gao**[1,2*], **Renjie Pi**[3*], **Jipeng Zhang**[3], **Jiacheng Ye**[2], **Wanjun Zhong**[1], **Yufei Wang**[1],
**Lanqing Hong**[1], **Jianhua Han**[1], **Hang Xu**[1], **Zhenguo Li**[1], **Lingpeng Kong**[2]
[1]Noah's Ark Lab    [2]The University of Hong Kong
[3]The Hong Kong University of Science and Technology
sumiler@connect.hku.hk, rpi@connect.ust.hk

## Abstract

Large language models (LLMs) have shown remarkable proficiency in human-level reasoning and generation capabilities, which encourages extensive research on their application in mathematical problem solving. However, current work has been largely focused on text-based mathematical problems, with limited investigation in problems involving multi-modal geometric information. Addressing this gap, we aim to enable LLMs to solve geometric problems by understanding image input. We first identify the limitations of current Multimodal Large Language Models (MLLMs) in this area: they struggle to accurately comprehend basic geometric elements and their relationships. To address these challenges, we leverage the inherent attribute of *logical structure compactness* in geometric figures, utilizing text-only Large Language Models (LLMs) to curate a comprehensive multimodal geometry dataset. This dataset, named Geo170K, contains more than 170K geometric image-caption and question-answer pairs. Utilizing the Geo170K dataset, we introduce G-LLaVA, a model that demonstrates exceptional performance in solving geometric problems. It significantly outperforms GPT4-V on the geometry task of MathVista benchmark with only 7B parameters.

## 1 Introduction

Large language models (LLMs) exhibit human-like proficiency in reasoning (Wang et al., 2022; Wei et al., 2022) and generation (Ouyang et al., 2022; Touvron et al., 2023), which encourages extensive research on their application in mathematical problem solving (Fu et al., 2023; Gou et al., 2023; Luo et al., 2023; Yue et al., 2023). These problems often require highly sophisticated and symbolic reasoning capabilities, which were considered impossible to solve before the era of LLMs.

It is an intuitive approach to use LLMs for mathematical reasoning problems presented in a textual form. However, a considerable portion of such problems requires understanding geometric information. Furthermore, even when certain problems may not explicitly involve geometric data, incorporating geometric-based methodologies often carries substantial practical implications (e.g., analytic number theory). With the advent of GPT4-V (OpenAI, 2023), Gemini (Google, 2023), and numerous multi-modal large language models (MLLMs) (Bai et al., 2023b; Dai et al., 2023; Gao et al., 2023b; Lai et al., 2023; Li et al., 2023; Liu et al., 2023; Zhu et al., 2023), recent work start to progressively look into employing MLLMs to tackle multi-modal geometric reasoning problems in mathematics (Lu et al., 2023; Yang et al., 2023).

However, we have observed that even the most advanced MLLMs still exhibit limitations in addressing geometric problems, due to challenges in accurately comprehending geometric figures. For instance, as demonstrated in Figure 1, GPT4-V often produces inaccurate descriptions for geometric figures. Specifically, the model struggles with understanding the relationships between fundamental elements like points and lines, and fail to accurately interpret elements such as the degrees of angles. The

---

* Equal Contribution.

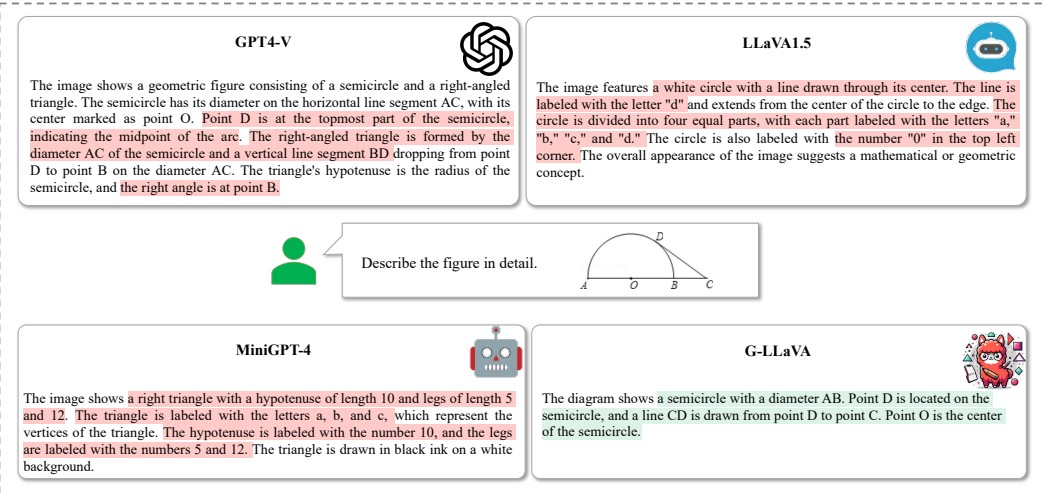

Figure 1: State-of-the-art MLLMs suffer severe hallucination on geometric figures, which greatly hinders their abilities on solving geometric problems. In contrast, after the first alignment phase, G-LLaVA demonstrates an enhanced ability in interpreting geometric figures compared to LLaVA.

underlying reason may be the fact that these MLLMs are predominantly trained with natural images, which differs significantly from geometric figures.

To address this issue, one direct and effective approach is to enhance current MLLMs by augmenting them with data containing high-quality descriptions of geometric information. However, a significant challenge is the limited size of the largest publicly available geometric problem dataset, which includes only a few thousand questions, each with a single solution. To collect new geometric datasets, it would introduce prohibitive expenses if human experts are hired to conduct labelling manually. On the other hand, automatic labelling with MLLMs is infeasible, since even the state-of-the-art MLLMs (e.g., GPT4-V) face challenges in comprehending geometric diagrams.

In this paper, we first highlight an inherent characteristic of geometric figures termed *logical structure compactness*. This property allows for the construction of high-quality multi-modal datasets using solely text-based LLMs such as ChatGPT. Specifically, logical structure is defined as the geometry elements and their relationships. With a compact logic structure, all the essential information for representing and reconstructing the geometric figure can be encompassed, which is owning to the concise nature of geometric elements (e.g., circles, triangles, lines). Therefore, upon acquiring expressions containing logic structures information, such as QA pair and logic form (Figure 2), which are prevalent in public geometric datasets, we can use them to represent the geometric figure and input them to text-only LLM, so that the LLM can interpret figure and produce high-quality annotations.

Specifically, we create two types of data: 1) *geometric cross-modal alignment data*, which aims at improving the MLLM's understanding of the geometric figures by aligning the image representations with their textual annotations, which also reduces hallucinations; 2) *geometric instruction data* that empowers the model with stronger geometric reasoning ability by covering more diverse geometric question scenarios. During data generation process, we leverage a series of geometric characteristics to enhance the quality, diversity, and scale of the training data (as shown in Figure 2). We term our generated dataset Geo170K, which contains around 60,000 geometric image-caption pairs and more than 110,000 question-answer pairs. This dataset is 28 times larger than GeoQA+ (the previous largest open-source dataset), greatly expanding the coverage of geometric problems.

With our collected Geo170K, we derive G-LLaVA, an MLLM capable of solving geometric problems, surpassing SOTA MLLMs by a large margin. Specifically, G-LLaVA-13B outperforms LLaVA-13B by 27.4 on the Geomtery Problem Solving (GPS) task on *testmini* split of MathVista (Lu et al., 2023). In addition, with only G-LLaVA-7B, it is able to surpass the powerful GPT4-V on the geometry problem solving questions. Our code, data, and models are publicly accessible at https://github.com/pipilurj/G-LLaVA.

## 2 RELATED WORK

**Geometry Problem Solving.** Geometry reasoning problem is a challenging visual mathematical reasoning problem. Early efforts by Alvin et al. (2017); Sachan and Xing (2017); Sachan et al. (2017); Seo et al. (2015) focused on creating datasets through manual efforts. More recent approaches have introduced enhanced methods and datasets, including Geometry3K (Lu et al., 2021), GeoQA (Chen et al., 2021), GeoQA+ (Cao and Xiao, 2022), UniGeo (Chen et al., 2022), UniMath (Liang et al., 2023), and SCA-GPS (Ning et al., 2023), aiming to improve both performance and explainability. However, the scale of current datasets remains limited, and the performance of traditional models in the multi-modal mathematical geometry problem domain remains markedly lower than progress has been made in text-based mathematical problems, particularly when compared to methods that utilize large language models for solving math word problems (Cobbe et al., 2021; Gou et al., 2023; Lu et al., 2023; Wei et al., 2022; Zhang et al., 2024b). More recently, significant progress has been made in multimodal mathematical reasoning. MathLLaVA (Shi et al., 2024) constructed a diverse dataset of 360k multimodal mathematical questions by utilizing GPT-4V to augment 40k seed examples from existing datasets. MultiMath (Peng et al., 2024) employed GPT-4O in multiple rounds of annotation to build the comprehensive MultiMath300K dataset, which spans K-12 levels and includes image captions and step-wise solutions. Additionally, MathPuMA (Zhuang et al., 2024) introduced a text-to-image rendering process that converts textual content into visual form, enhancing the diversity of image sources. While these works have significantly advanced the field, our approach differs by primarily focusing on augmenting datasets from the textual side to enhance the MLLM's understanding of geometry. Without relying on expensive models like GPT-4V or GPT-4o for data annotation, we generate multimodal data using text-only LLMs. This text-centric augmentation serves as a complementary method to image-centric approaches, offering greater scalability and cost-effectiveness. By leveraging text-only LLMs, we demonstrate that substantial improvements in geometric reasoning can be achieved, highlighting an alternative pathway to advancing MLLMs without extensive reliance on visual data augmentation.

**Data Generation via LLM.** Bootstrapping data from pre-trained models has long been an active area of research. Meng et al. (2022); Ye et al. (2022a;b) generates training data using pre-trained language models such as GPT-2 for classification tasks. Gao et al. (2023a) improves the quality of generated dataset via bi-level approach. Recently, automatic data generation becomes more ubiquitous with the advent of powerful LLMs such as ChatGPT, a line of recent works utilize LLM-generated data to perform instruction tuning (Huang et al., 2024; Li et al., 2024; Peng et al., 2023; Pi et al., 2024b; Taori et al., 2023; Wang et al., 2023; Zhang et al., 2024a).

**Multi-Modal Large Language Model.** Recent years have witnessed transformative advancements in the development of large language models (LLMs), characterized by a series of pioneering studies (Bai et al., 2022; Brown et al., 2020; Chowdhery et al., 2022; Hoffmann et al., 2022; Ouyang et al., 2022; Scao et al., 2022; Smith et al., 2022; Touvron et al., 2023). These breakthroughs have significantly elevated the capabilities of language understanding and generation, showcasing near-human proficiency across diverse tasks. Concurrently, the success of LLMs has inspired explorations into vision-language interaction, leading to the emergence of multi-modal large language models (MLLMs) (Bai et al., 2023b; Dai et al., 2023; Dong et al., 2024; Gao et al., 2024a;b; Li et al., 2023; Liu et al., 2023; 2024; Pi et al., 2023a;b; 2024a; Su et al., 2023; Zhu et al., 2023). These models have exhibited remarkable capabilities in synthesizing detailed descriptions and engaging in dialogue based on visual inputs. However, we observe that even the state-of-the-art MLLMs face challenges in resolving geometric problems using diagrams and figures.

## 3 OBSERVATION

We observe that state-of-the-art (SOTA) MLLMs, although being adept at understanding natural images, have difficulty in comprehending simple geometric figures. In Figure 1, we demonstrate that the descriptions from SOTA MLLMs for geometric figures suffer from severe hallucination. This inadequacy in interpreting geometric diagrams may be one of the major causes of the failure in solving geometric problems.

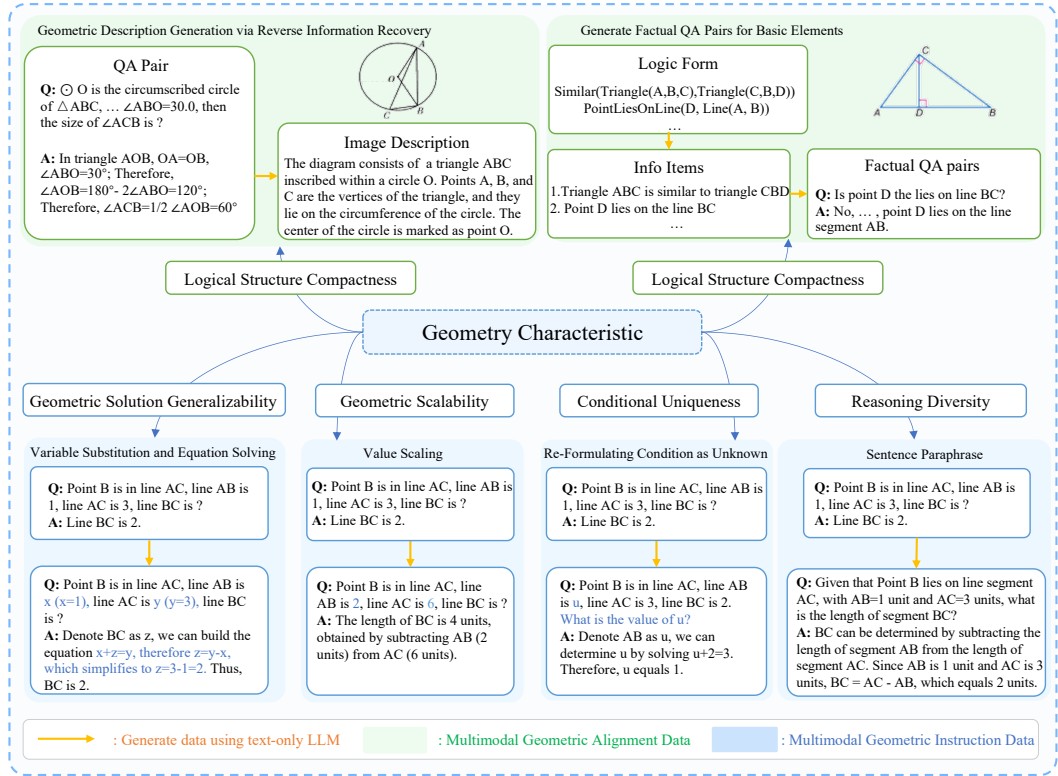

Figure 2: Framework of multi-modal geometric data generation using geometry characteristics.

Specifically, we find GPT4-V has difficulty understanding the basic elements, such as points, lines and angles (e.g., angle B in Figure 1). Furthermore, it struggles to comprehend the spatial relationships between the basic elements. On the other hand, open-source MLLMs like LLaVA1.5 and MiniGPT4 (7B) face even greater challenges in accurately identifying the geometric shapes. This inadequacy in geometric interpretation may be a major cause of the failure in solving geometric problems.

To address this issue, we point out that geometric figures enjoy an appealing property, which we refer to as *logical structure compactness*. Specifically, logic structure is defined as the geometric elements (shapes) and their spatial relationships. Due to the simplicity of geometric elements such as circles, triangles, and lines, geometric figures can be accurately represented by leveraging logical structure information. This stands in contrast to natural images, where such precise reconstruction is not possible. In publicly available geometric datasets, the annotations for question-answer pairs and the logic form representation are precise expressions that capture the underlying key logical structures inherent in geometry figures (see Figure 2). By leveraging on this characteristic, we can utilize these

Table 1: Geometric diagram description generation via information recovery. The description is generated based on the textual QA pair. The upper section shows the QA pair employed to instruct text-only ChatGPT, while the lower section ( in blue ) shows the responses generated by ChatGPT.

---

**Geometric Description Generation via Inverse Information Recovery**

**QA Pair:** Question: As shown in the figure, circle O is the circumscribed circle of triangle ABC, and it is known that angle ABO = 30.0, then the size of angle ACB is ()
Answer: In triangle AOB, OA=OB, angle ABO=30°; Therefore, angle AOB=180°- 2 angle ABO=120°; Therefore, angle ACB=1/2 angle AOB=60°

---

**Diagram Description:**
The diagram consists of a triangle ABC inscribed within a circle, where the circle is denoted as circle O. Points A, B, and C are the vertices of the triangle, and they all lie on the circumference of the circle. The center of the circle is marked as point O.

compact representations instead of the original geometric figure as input to text-only LLM, enabling the LLM to comprehend the geometric figure and generate annotations with high quality.

## 4 GEOMETRIC DATA GENERATION

The key limitations of existing datasets (Cao and Xiao, 2022; Chen et al., 2021; 2022) are threefold: (1) the limited size; (2) the absence of detailed descriptions for geometric images; and (3) the lack of diversity in problem-solving methodologies and reasoning paths. To address these challenges, we leverage the logic structure compactness of geometric figures to construct a multi-modal geometry dataset, building upon existing datasets. This dataset includes two parts: an alignment dataset and an instruction-tuning dataset.

### 4.1 GEOMETRIC CROSS-MODAL ALIGNMENT DATA

**Geometric Image Description Generation.** Inspired by previous works on MLLMs (Liu et al., 2023; Zhu et al., 2023), detailed image-description pairs are crucial for training MLLMs to comprehend image contents. However, there is a lack of image-text paired dataset for geometry problems.

---

**Factual QA Pairs for Basic Elements**

**Logic Form:**
```
Similar(Triangle(A,B,C),Triangle(C,B,D))
Triangle(A,B,C)
Triangle(A,C,B)
Equals(LengthOf(Line(C, B)), 5)
Equals(LengthOf(Line(C, D)), 3)
Equals(LengthOf(Line(B, D)), 4)
PointLiesOnLine(D, Line(A, B))
Perpendicular(Line(A, C), Line(C, B))
Perpendicular(Line(C, D), Line(A, D))
```

---

*Phase-1 Generation*

**Info Items:**
The geometric figure have the following information:
Info 1: The figure represents a triangle with vertices A, B, and C.
Info 2: Triangle ABC is similar to triangle CBD.
Info 3: Point D lies on the line segment AB.
Info 4: Line segment AC is perpendicular to line segment CB.
...

**Summary:**
The figure depicts a triangle ABC and triangle CBD, which is similar to triangle ABC. Additionally, triangle ACB is also present in the figure. The lengths of the segments CB, CD, and BD are 5, 3, and 4 units respectively. Point D lies on the line segment AB, and perpendicularity is indicated between the line segments AC and CB, as well as between the line segments CD and AD.

---

*Phase-2 Generation*

**QA pairs:**
1. Question: Does the diagram include a triangle formed by vertices A, B, and D?
Answer: **No**, the diagram does not include a triangle formed by vertices A, B, and D. The triangles present are ABC and CBD, and the point D lies on the line segment AB.
2. Question: Does the diagram contain a triangle with vertices A, C, and B?
Answer: **Yes**, the diagram contains triangle ACB, which is the same as triangle ABC.
3. Question: Is point D the lies on line BC?
Answer: **No**, point D is not on lines BC. The point D lies on the line segment AB.
4. Question: Is line segment AC perpendicular to line segment CB?
Answer: **Yes**, line segment AC is perpendicular to line segment CB.

---

Table 2: Geometric image description and factual QA pairs for understanding basic elements. The sections in blue display the responses generated by ChatGPT.

To close this gap, we propose a novel approach that leverages the unique logic structure compactness property to generate comprehensive descriptions effectively from labeled question-answer (QA) pairs, as illustrated in Table 1. In particular, we leverage the strong comprehension abilities of text-only ChatGPT 3.5 to generate image captions given human-labeled textual question-answer (QA) pairs that contain essential information of the logic structure. This process can be viewed as a type of inverse information recovery. The generated descriptions are paired with existing images to form the image-description datasets.

**Factual QA Pairs for Basic Elements.** We generate QA pairs to facilitate the comprehension of geometric diagrams, focusing on basic geometric elements. The process begins with the interpretation of human-labeled logical forms that capture the entire logic structures of geometric figures on Geometry3k (Lu et al., 2021). In the Phase-1 generation, we employ ChatGPT to convert these logical forms into textual descriptions that cover various geometric elements (e.g., shapes, lines, and points) and their relationships. In Phase-2 generation, we use ChatGPT to produce factual QA pairs given both the logic form and the result from Phase-1. The whole process is shown in Table 2.

The QA pairs may explore the presence of certain geometric elements (e.g., "Are there triangular shapes in the diagram?") or check the accuracy of the relationships described (e.g., "Is point D the lies on line BC?"). The answers are "Yes" or "No", along with the corresponding reasoning pathway. This method enables the model to comprehend geometric concepts and interpret the details in geometric diagrams accurately (Table 2). The detailed prompts are shown in Appendix D.

**Remark.** The alignment data is used during the first-stage cross-modal alignment to enhance the model's ability to understand geometric diagrams. The effectiveness is shown in Figure 1.

Table 3: The original example.

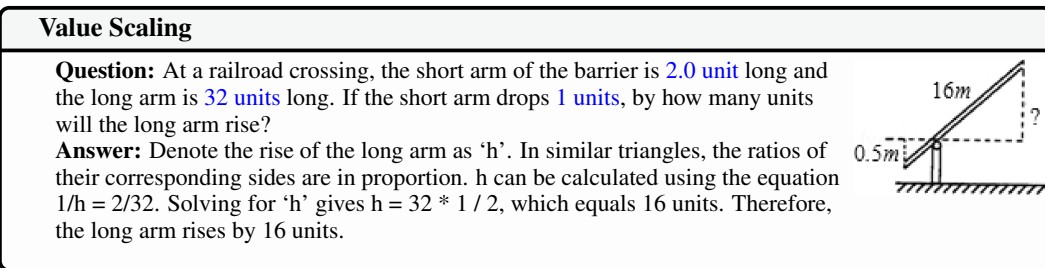

| Original Example |
| --- |
| **Question:** As shown in the figure, the length of the short arm of the railing at the railway crossing is 1.0 and the length of the long arm is 16.0. When the end of the short arm drops by 0.5, the end of the long arm rises ()
**Answer:** By the property of similar triangles, the lengths of corresponding sides are proportional. Since the long arm is 16 times longer than the short arm, the height it rises is 16 times the height the short arm drops, which is 0.5 multiplied by 16, resulting in a rise of 8 meters. |

Table 4: Synthetic example via value scaling.

| Value Scaling |
| --- |
| **Question:** At a railroad crossing, the short arm of the barrier is 2.0 unit long and the long arm is 32 units long. If the short arm drops 1 units, by how many units will the long arm rise?
**Answer:** Denote the rise of the long arm as 'h'. In similar triangles, the ratios of their corresponding sides are in proportion. h can be calculated using the equation $1/h = 2/32$. Solving for 'h' gives h = 32 * 1 / 2, which equals 16 units. Therefore, the long arm rises by 16 units. |

## 4.2 GEOMETRIC CROSS-MODAL INSTRUCTION DATA

To improve the model's ability to understand user instructions and provide solutions to geometric problems, we further curate an instruction-tuning dataset based on a series of geometric characteristics, which is enriched with diverse question and reasoning paths.

**Value Scaling (VS).** We note that geometric QA pairs remain correct if all the lengths in a geometric problem are scaled simultaneously by the same factor, which we refer to as *geometric scalability*. However, note that this is not the case for quantities like angles. Based on this, we augment length-related problems by scaling all the length values in the QA pairs via the same factor (as shown in

Table 5: Synthetic example via replacing values to unknown variables and solving the equation.

**Variable Substitution and Equation Solving**

**Question**: Given a railway crossing barrier with two arms, where the shorter arm has a length of 'a' units (a = 1.0) and the longer arm has a length of 'b' units (b = 16.0), determine the elevation 'y' units of the longer arm's end when the shorter arm descends by 'd' units (d = 0.5). Express your answer as a function of 'a', 'b', and 'd'.

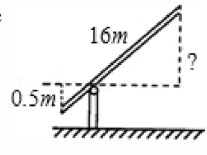

**Answer:** To find the elevation 'y' of the long arm's end, we apply the property of similar triangles and establish the equation $\frac{d}{y} = \frac{a}{b}$. Substituting the given values, we get $\frac{0.5}{y} = \frac{1.0}{16.0}$. Solving for 'y' leads to $y = \frac{b \times d}{a}$, which simplifies to $y = \frac{16.0 \times 0.5}{1.0}$. Therefore, 'y' equals 8.0 units, indicating that the long arm rises by 8.0 units when the short arm descends by 0.5 units.

Table 6: Synthetic example via re-formulating condition as unknown.

**Re-Formulating Condition as Unknown**

**Question:** At a railroad crossing, the short arm of the barrier is u unit long and the long arm is 16.0 units long. When the end of the short arm drops by 0.5, the end of the long arm rises 8 units. What is the value of unknown variable u?

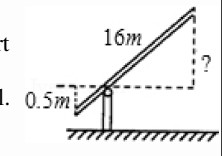

**Answer:** Denote the short arm of the barrier as variable u. By the property of similar triangles, we can determine u by solving the equation 0.5/8 = u/16. Therefore, u equals 1.

Table 7: Synthetic example via sentence paraphrase.

**Sentence Paraphrase**

**Question:** In the illustration, the railing at the railway crossing has a short arm measuring 1.0 unit in length and a long arm measuring 16.0 units. When the short arm drops by 0.5 units, what is the corresponding rise in the long arm?

**Answer:** The triangles are similar, and their corresponding sides are proportional. The long arm is 16 times longer than the short arm, resulting in an 8-meter rise when the short arm drops by 0.5 meters.

Table 4). When different scalings are applied, the LLM becomes more flexible in handling diverse numerical inputs, which aids in refining the model's reasoning capabilities and generalizability.

**Equation Solving (ES).** As shown in Table 5, we replace the specific values in the original QA pairs with unknown variables and prompt the LLM to construct the solution by solving equation. Such data helps the MLLM generalize its understanding of the problem, which enables it to apply the similar reasoning and solution steps to different scenarios. The abstraction of the problem by using variables and equations helps the MLLM focus on the underlying mathematical concepts and relationships, rather than overfitting to specific values.

**Re-Formulating Condition as Unknown (RCU).** The inherent logical structures within the question-answer pairs enable the reconstruction of the geometric figures based on the provided information. Consequently, it becomes possible to interchange the known and unknown variables, wherein the condition stated in the question can be considered as the variable, while the unknown variable can be viewed as the condition (Weng et al., 2023; Yu et al., 2023). In this way, the MLLM is repeatedly exposed to the relationships between the elements. This reinforcement helps the model learn the dependencies and connections between different elements in the geometric problem. An example is shown in Table 6.

**Sentence Paraphrase (SP).** As shown in Table 7, we paraphrase both the questions and answers, exposing the LLM to broader range of phrasing and language variations. This makes the model more robust in understanding questions in various forms and generating diverse reasoning paths.

**Data Correctness Verification.** We implement a verification process to ensure the correctness of the reasoning paths when generating new question-answer pairs. Specifically, we require Chat-GPT to finish the answer in a specific pattern, marked as "Result:[RESULT]". Subsequently, we employ regular expression to extract the answers and compare them to the expected true answers. To ensure correctness of our data, only the generated QA samples that are associated with the correct answers are retained. For example, in the VS approach, we verify whether the answer is equal to the original answer multiplied by the scaling factor, and in the RCU approach, we check whether the generated answer matches the value provided in the condition.

## 5 EXPERIMENTS

### 5.1 SETUP

**Dataset.** We generate the alignment data and instruction data utilizing training set of GeoQA+ (Cao and Xiao, 2022) and Geometry3K (Lu et al., 2021). More specifically, the factual QA pairs in the alignment data are generated using Geometry3K, which contain human-annotated logical forms. We evaluate G-LLaVA on the geometry problems solving (GPS) task (*testmini* split) of MathVista (Lu et al., 2023) and test set of GeoQA. Note that the GPS task in MathVista *testmini* set is collected from GeoQA, Geometry3K, GeoQA+, and GEOS (Seo et al., 2015). To verify the generalization ability of G-LLaVA, we also conduct evaluation on the newly introduced MathVerse benchmark (Zhang et al., 2024b).

**Model.** We develop our model based on LLaVA (Liu et al., 2023). More specifically,s we utilize LLAMA-2 (Touvron et al., 2023) as the language model and employ the pretrained vision transformer Radford et al. (2021) as the vision encoder (ViT). We conduct experiments with both 7B and 13B LLMs. We employ ChatGPT (gpt-3.5-turbo-0613) for data generation. The detailed description of our prompts is provided in Appendix D.

**Training.** We train our model in two phases, namely 1) geometric visual-language alignment, and 2) geometric instruction tuning. Each phase is conducted with the corresponding dataset, respectively. In both phases, we leverage the conventional language modeling loss. More implementation details are provided in Appendix E.

**Evaluation Metric.** We use accuracy as the evaluation metric. Several prior studies (Cao and Xiao, 2022; Chen et al., 2021; 2022) report results using Top-10 accuracy (generating 10 sequences and se-

Table 8: Results on *testmini* set of Math-Vista (Lu et al., 2023) on geometry problem solving (GPS) . For input, $Q$ represents for question, $I$ represents for image, $I_c$ represents for image caption generated by Bard, and $I_t$ represents fo OCR text detected in the image. Baseline results are obtained from Lu et al. (2023). Human performance and the results surpassing human performance are highlighted in grey. Our results are highlighted in blue .

| Model | Input | Accuracy (%) |
|---|---|---|
| *Heuristics Baseline* | | |
| Random Chance | - | 21.6 |
| Frequent Guess | - | 34.1 |
| Human | $Q, I$ | 48.4 |
| *Close Source Model* | | |
| *Text-Only LLMs* | | |
| Zero-shot ChatGPT | $Q$ | 26.9 |
| Zero-shot GPT-4 | $Q$ | 37.0 |
| 2-shot CoT Claude-2 | $Q$ | 29.8 |
| 2-shot CoT ChatGPT | $Q$ | 36.5 |
| 2-shot CoT GPT-4 | $Q$ | 44.7 |
| 2-shot PoT ChatGPT | $Q$ | 30.8 |
| 2-shot PoT GPT-4 | $Q$ | 33.2 |
| *Visual-Augmented LLMs* | | |
| 2-shot CoT Claude-2 | $Q, I_c, I_t$ | 31.7 |
| 2-shot CoT ChatGPT | $Q, I_c, I_t$ | 29.3 |
| 2-shot CoT GPT-4 | $Q, I_c, I_t$ | 31.7 |
| 2-shot PoT ChatGPT | $Q, I_c, I_t$ | 26.4 |
| 2-shot PoT GPT-4 | $Q, I_c, I_t$ | 39.4 |
| *Multimodal LLMs* | | |
| Multimodal Bard | $Q, I$ | 47.1 |
| SPHINX (V1) | $Q, I$ | 23.1 |
| SPHINX (V2) | $Q, I$ | 16.4 |
| Gemini Nano 1 | $Q, I$ | 21.6 |
| Gemini Nano 2 | $Q, I$ | 23.6 |
| Gemini Pro | $Q, I$ | 40.4 |
| Gemini Ultra | $Q, I$ | 56.3 |
| GPT4-V | $Q, I$ | 50.5 |
| *Open Source Model* | | |
| IDEFICS (9B-Instruct) | $Q, I$ | 21.1 |
| mPLUG-Owl (LLaMA-7B) | $Q, I$ | 23.6 |
| MiniGPT4 (LLaMA-2-7B) | $Q, I$ | 26.0 |
| LLaMA-Adapter-V2 (7B) | $Q, I$ | 25.5 |
| LLaVAR | $Q, I$ | 25.0 |
| InstructBLIP (Vicuna-7B) | $Q, I$ | 20.7 |
| LLaVA (LLaMA-2-13B) | $Q, I$ | 29.3 |
| G-LLaVA-7B | $Q, I$ | 53.4 |
| **G-LLaVA-13B** | $Q, I$ | **56.7** |

lecting the first sequence that successfully addresses the problem as the prediction). In our paper, our experimental results directly report Top-1 accuracy. More details are attached to Appendix E.

### 5.2 EXPERIMENTAL RESULTS

**Main Experiments.** We compare MLLMs on *testmini* split of MathVista (Lu et al., 2023) benchmark on Table 8. The results shows that, geometric cross-modal alignment and instructing tuning

on our dataset is effective in improving MLLMs' geometric problem solving ability. Our specific in-domain model G-LLaVA-7B can even surpass the strong GPT4-V on geometric problems.

**Comparison with Conventional Methods.** We additionally compare our method with conventional SOTA methods in geometry problem solving domain. As illustrated in Table 9, our method demonstrates a notable improvement in Top-1 accuracy over the existing SOTA techniques. Moreover, our model's top-1 accuracy outperforms the baselines' top-10 accuracy, demonstrating a significant improvement in predictive precision.

**Effectiveness of Cross-Modal Alignment.** To evaluate the alignment phase's effectiveness, we conducted experiments with and without geometric alignment phase in Table 10. The results suggest that the alignment phase enhances the model's ability to interpret images, which is also illustrated by the qualitative result in Figure 1. We also observe that releasing the LLM during the alignment phase leads to more hallucinations, which consequently degrade its performance.

**Gain Introduced by Different Types of Augmentation.** We demonstrate the performance gains introduced by each augmentation strategy, as well as the cumulative number of training data. As shown in Table 11, each of our proposed augmentation strategies is able to boost the model performance. Note that as the accuracy grows higher, the extra gain becomes more difficult.

**Performance Across Problem Difficulties.** We perform experiments on problems with different difficulties, as shown in Table 12. Specifically, OP represents the number of "operations", or reasoning steps that needs to be taken for solving the problem. The results verify that our G-LLaVA consistently outperforms baseline models by a large margin across various difficulty levels.

Table 9: Comparison of model performance with traditional methods on GeoQA.

| Model | Input | Accuracy (%) |
|---|---|---|
| Random Chance | - | 25.0 |
| Frequent Guess | - | 32.1 |
| *Top-10 Accuracy* | | |
| NGS (Chen et al., 2021) | $Q, I$ | 56.9 |
| DPE-GPS (Cao and Xiao, 2022) | $Q, I$ | 62.7 |
| SCA-GPS (Ning et al., 2023) | $Q, I$ | 64.1 |
| *Top-1 Accuracy* | | |
| Geoformer (Chen et al., 2022) | $Q, I$ | 46.8 |
| UniMath (Liang et al., 2023) | $Q, I$ | 50.0 |
| **G-LLaVA-7B** | $Q, I$ | **64.2** |
| **G-LLaVA-13B** | $Q, I$ | **67.0** |

Table 10: Effectiveness of alignment in the pre-training phase. Top-1 accuracy is reported. AL stands for first-stage cross-modal alignment.

| Model | Input | Accuracy |
|---|---|---|
| Random Chance | - | 25.0 |
| Frequent Guess | - | 32.1 |
| Geoformer (Chen et al., 2022) | $Q, I$ | 46.8 |
| UniMath (Liang et al., 2023) | $Q, I$ | 50.0 |
| LLaVA-7B | $Q, I$ | 18.7 |
| + SFT | $Q, I$ | 62.8 |
| + AL (tunable LLM ) + SFT | $Q, I$ | **60.1** |
| + AL (fixed LLM ) + SFT | $Q, I$ | **64.2** |

Table 11: The impact of incorporating data augmentation on GeoQA. For this ablation study, the training set covers the full GeoQA+ train set.

| Training Data | Data Size | Accuracy |
|---|---|---|
| GeoQA+ train set | 6027 | 48.2 |
| w/ SP | 75,982 | 56.1 |
| w/ ES | 100,411 | 59.4 |
| w/ VS | 109,787 | 62.3 |
| w/ RCU | 119,563 | 65.1 |

**Performance Across Different Types of Questions.** We compare G-LLaVA with the baselines models on problems with different type of questions, as shown in Table 13. The results suggest that G-LLaVA performs better than the baseline models in various geometric problems such as angle, length, and area problems.

**Generalization Ability.** We also conduct experiment on the newly-introduced benchmark Math-Verse (Zhang et al., 2024b). For ease of evaluation, we conduct evaluation on the multiple-choice (MC) questions. The full results of the *testmini* is collected from Zhang et al. (2024b). According to results in Table 14, G-LLaVA significantly surpasses other open-source MLLMs in the Mathverse

Table 12: Performance of different difficulty problems on GeoQA.

| Model | OP=1 | OP=2 | OP=3 | OP>=4 | Total |
|---|---|---|---|---|---|
| LLaVA-7B | 16.8 | 20.9 | 15.5 | 22.9 | 18.7 |
| LLaVA-13B | 19.1 | 21.3 | 18.5 | 24.6 | 20.3 |
| **G-LLaVA-7B** | **77.5** | **60.8** | **54.8** | **40.9** | **64.2** |
| **G-LLaVA-13B** | **79.0** | **64.9** | **55.5** | **49.1** | **67.0** |

Table 13: Performance on different types of questions on GeoQA.

| Model | Angel | Length | Area | Other | Total |
|---|---|---|---|---|---|
| LLaVA-7B | 16.1 | 22.2 | 17.0 | 14.3 | 18.7 |
| LLaVA-13B | 17.5 | 23.0 | 25.5 | 28.6 | 20.3 |
| **G-LLaVA-7B** | **70.7** | **56.5** | **55.3** | **42.9** | **64.2** |
| **G-LLaVA-13B** | **71.5** | **61.1** | **63.8** | **57.1** | **67.0** |

Table 14: Model performance on MathVerse *testmini* set.

| Model | All | Text Dominant | Text Lite | Text Only | Vision Intensive | Vision Dominant | Vision Only |
|---|---|---|---|---|---|---|---|
| | | *testmini* | | | | | |
| LLaMA-Adapter (Gao et al., 2023c) | 5.7 | 6.2 | 5.9 | 2.7 | 6.1 | 4.2 | 6.1 |
| ImageBind-LLM (Han et al., 2023) | 9.3 | 11.4 | 11.3 | 11.7 | 8.9 | 11.2 | 3.4 |
| mPLUG-Owl2 (Ye et al., 2023) | 4.6 | 6.6 | 6.3 | 6.1 | 6.3 | 5.6 | 4.9 |
| MiniGPT-v2 (Chen et al., 2023a) | 11.0 | 12.1 | 12.0 | 11.7 | 13.1 | 10.3 | 7.4 |
| SPHINX-Plus (Gao et al., 2024b) | 12.2 | 13.9 | 11.6 | 14.9 | 11.6 | 13.5 | 10.4 |
| LLaVA-NeXT (Liu et al., 2024) | 10.3 | 12.8 | 12.0 | 9.9 | 10.7 | 9.7 | 6.3 |
| ShareGPT4V (Chen et al., 2023b) | 13.1 | 16.2 | 16.2 | 6.6 | 15.5 | 13.8 | 3.7 |
| Qwen-VL-Plus (Bai et al., 2023a) | 11.8 | 15.7 | 11.1 | 14.5 | 9.0 | 13.0 | **10.0** |
| LLaVA-7B (Liu et al., 2023) | 7.6 | 8.8 | 7.6 | 11.5 | 7.4 | 7.4 | 6.9 |
| G-LLaVA-7B | **16.6** | **20.9** | **20.7** | **21.1** | **17.2** | **14.6** | 9.4 |
| | | *testmini-MC* | | | | | |
| LLaVA-7B | 11.0 | 12.8 | 13.1 | 12.8 | 14.2 | 13.3 | 0.0 |
| LLaVA-13B | 16.7 | 22.7 | 21.1 | 19.0 | 18.1 | 19.3 | 0.0 |
| **G-LLaVA-7B** | **34.7** | **42.7** | **36.7** | **42.4** | **31.2** | **28.2** | **27.3** |
| **G-LLaVA-13B** | **36.3** | **43.1** | **37.4** | **45.2** | **32.3** | **29.4** | **30.3** |

benchmark, demonstrating its generalization ability. Our model also outperforms LLaVA-7B (our base model) by a margin of 9.0, which highlights the quality of our annotated dataset. Moreover, G-LLaVA not only shows substantial improvements in text-dominant, text-lite, text-only, and vision-intensive categories, but also exhibits advancements in vision-dominant and vision-only categories. This verifies G-LLaVA's capability to interpret geometric figures and solve problems effectively.

Additionally, we show the results of sub-tasks on MathVista-GPS multiple-choice questions for easier evaluation in Table 15, which includes GeoQA+, UniGeo, Geometry3K, and GEOS. We observed a significant performance improvement from 23.5% to 47.1% in the unseen GEOS task. This advancement confirms that our dataset improves the MLLM's ability to interpret geometric figures and enhances its generalization capabilities.

Table 15: Subtask performance of MathVista on geometry problem-solving (MC).

| Model | All | GeoQA+ | UniGeo | Geometry3K | GEOS |
|---|---|---|---|---|---|
| LLaVA-7B | 15.8 | 6.5 | 24.2 | 14.5 | 23.5 |
| **G-LLaVA-7B** | **55.7** | **56.5** | **72.6** | **40.3** | **47.1** |

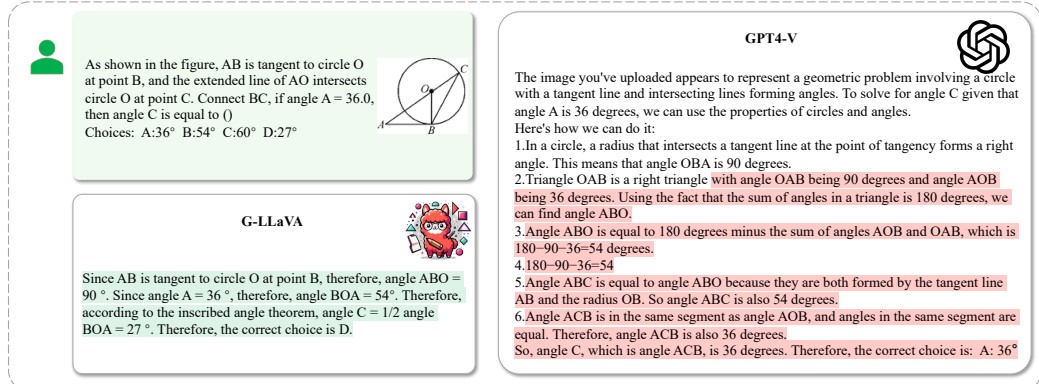

Figure 3: Visualization of problem solving by GPT-4-V and G-LLaVA.

## 6 CONCLUSION

This paper attempts to address the limitations of MLLMs in solving geometric problems. Taking advantage of geometric logic structure compactness, we propose several strategies to enrich the multi-modal geometric data by leveraging text-only LLMs, resulting in the generated dataset, Geo170K. With this dataset, our G-LLaVA outperforms GPT-4-V on the geometric task (GPS) of MathVista, with as few as 7B parameters. We hope our work provides new insights on improving multi-modal LLMs' ability to solve geometric problems.

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

## A    REPRODUCIBILITY STATEMENT

We develop an innovative, automatic, and scalable data-generation method that leverages a text-only GPT to create multimodal data. Detailed instructions for data generation are available in Appendix D. We employed a standardized training protocol to facilitate fair comparisons, with implementation details provided in Appendix E. Our code, data, and models are publicly accessible at https://github.com/pipilurj/G-LLaVA.

## B    ETHICAL IMPACT

Our work aims to enhance the geometric problem-solving ability of MLLMs without introducing any additional ethical concerns or resolving existing ones.

## C    ADDITIONAL EXAMPLES

### C.1    FULL EXAMPLE FOR FACTUAL QA PAIRS GENERATION

The factual QA pairs generation process consists of two phases: 1) Translating the human-labelled logic form into detailed information items and a summary of the diagram description. 2) Generating factual QA pairs based on the provided information and summary. We utilize ChatGPT-3.5 as the LLM for data generation. The full example is shown in Table 17. The detailed prompts are shown in Table 20 and Table 21.

### C.2    ADDTITONAL EXAMPLE FOR EQUATION SOLVING(ES) STRATEGY

We attach additional synthetic examples by using Equation Solving(ES) strategy in Table 18.

## D    PROMPTS

### D.1    PROMPT FOR GENERATING GEOMETRIC CROSS-MODAL ALIGNMENT DATA

The detailed prompt for generating geometrical image description is shown in Table 19, and the prompts for generating factual QA pairs are shown in Table 20 and Table 21.

### D.2    PROMPTS FOR GENERATING INSTRUCTION DATA

The prompt for Equation Solving (ES), Value Scaling (VS), Re-Formulating Condition as Unknown (RCU), and Sentence Paraphrase (SP) are shown in Table 22, Table 23, Table 24, and Table 25, respectively. "[The Original Full Information]" in Table 24 denotes the original question with answers completed based on the ground truth.

## E    MORE EXPERIMENTAL DETAILS

### E.1    MODEL ARCHITECTURE

We utilize the LLaVA (Liu et al., 2023) architecture for our model. The model mainly consists of a large language model (LLM) such as Vicuna Chiang et al. (2023), a pre-trained vision transformer Radford et al. (2021) (ViT) as image encoder. In addition, a projection layer is required to map the visual features from the image encoder to the same dimension as the LLM.

During inference, given an image and a textual instruction, the image encoder first extracts the visual tokens from the image, which are then mapped to the dimension of LLM's embedding space via the projection layer. Then, the mapped image features are concatenated with text embeddings to serve as the input to the LLM. Subsequently, the LLM begins to perform next-token-generation.

### E.2    MODEL TRAINING

We train our G-LLaVA in two phases, namely 1) geometric visual-language alignment, and 2) geometric instruction tuning. In both phases, we leverage the conventional language modeling loss, which can be formulated as follows:

$$\mathcal{L}(S_{\text{tar}}, S_{\text{in}}, I) = -\sum_{t=1}^{L} \log p \left[ S_{\text{tar}}^t \mid \mathcal{F}(s_{\text{tar}}^{(<t)}, S_{\text{in}}, I) \right] \tag{1}$$

where $\mathcal{F}$ represents the model. $I$ is the geometric figure; $S_{tar}$ and $S_{in}$ represent the target and input sentences, respectively; $S_{tar}^t$ denotes the $t^{th}$ token of the target output, and $L$ stands for its length.

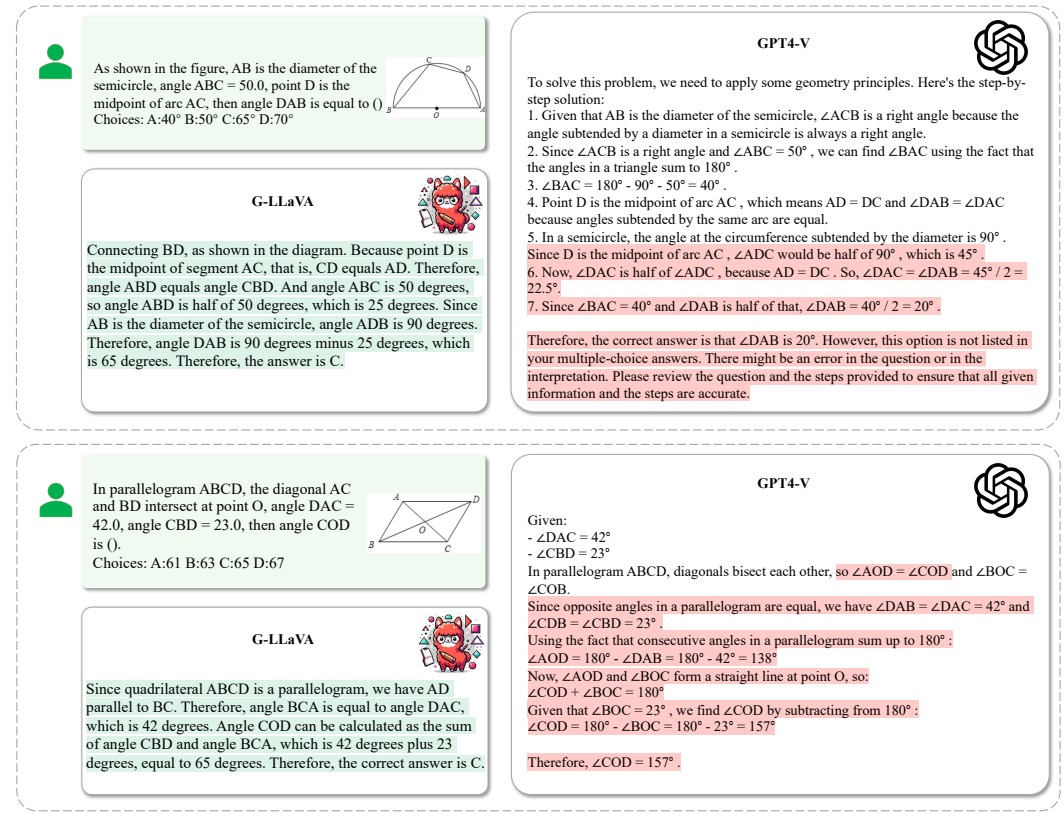

Figure 4: Other cases of geometric problem solving by GPT-4-V and G-LLaVA.

### E.3 SETUP

**Dataset.** We generate the alignment data and instruction data utilizing training set of GeoQA+ (Cao and Xiao, 2022) and Geometry3K (Lu et al., 2021). More specifically, the contrastive question-answer (QA) pairs in the alignment data are generated using Geometry3K, which features human-labeled logical forms. Note that GeoQA+ covers the training set of GeoQA (Chen et al., 2021), and share the same val/test set as GeoQA (Chen et al., 2021). More details of data split on GeoQA and GeoQA+ is listed in Table 16. Our approach results in 60K alignment data samples, and more than 110K instruction data samples.

We compare MLLMs on the geometry problems solving (GPS) task on *testmini* split of MathVista (Lu et al., 2023), a popular benchmark for visual math reasoning. This benchmark has assessed several leading MLLMs such as GPT4-V and Gemini. We also compare our model with traditional in-domain models on the test split of GeoQA following (Liang et al., 2023). Note that the GPS task in MathVista *testmini* set is collected from four source datasets: GeoQA (Chen et al., 2021), Geometry3K (Lu et al., 2021), GeoQA+ (Cao and Xiao, 2022), and GEOS (Seo et al., 2015).

**Implementation Details.** We employ ChatGPT (gpt-3.5-turbo-0613) for data generation. The LLM part of G-LLaVA utilizes LLAMA-2 (Touvron et al., 2023) as the language model and employ the pretrained vision transformer Radford et al. (2021) as the vision encoder. We conduct experiments with both 7B and 13B LLMs.

For model training, as we found the *testmini* split of MathVista contains some examples of Mix-train.pk of GeoQA+, we remove the samples that also appears in *testmini* split of MathVista. During training, the learning rate is set to $3e^{-5}$. We expand the images into squares during training, where the extended background color is set to white. For image augmentation, we set the maximum translation distance to 0.25 of the length of longer side. If not otherwise specified, the models are trained for 1

epoch for cross-modal alignment and 2 epochs for instruction tuning, respectively. And the batch sizes are set to 6 and 32 per GPUs, respectively. For training G-LLaVA-7B, each run requires 10 hours on 8 A40 GPUs (48G of memory).

# F  ADDITIONAL CASES OF GEOMETRY PROBLEM SOLVING

We provide additional cases of geometry problem solving using different models in Figure 4. The results show that G-LLaVA can provide accurate reasoning paths and answers.

| Dataset | Train | Validation | Test |
|---|---|---|---|
| GeoQA+ (Cao and Xiao, 2022) | 6027 | 745 | 754 |
| GeoQA (Chen et al., 2021) | 3499 | 745 | 754 |

Table 16: Data Split of GeoQA and GeoQA+.

---

**Factual QA Pairs for Basic Elements**

**Logic Form:**
```
Similar(Triangle(A,B,C),Triangle(C,B,D))
Triangle(A,B,C)
Triangle(A,C,B)
Equals(LengthOf(Line(C, B)), 5)
Equals(LengthOf(Line(C, D)), 3)
Equals(LengthOf(Line(B, D)), 4)
PointLiesOnLine(D, Line(A, B))
Perpendicular(Line(A, C), Line(C, B))
Perpendicular(Line(C, D), Line(A, D))
```

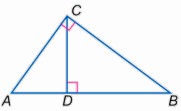

---

*Phase-1 Generation*

**Info Items:**
The geometric figure have the following information:
Info 1: The figure represents a triangle with vertices A, B, and C.
Info 2: Triangle ABC is similar to triangle CBD.
Info 3: Point D lies on the line segment AB.
Info 4: Line segment AC is perpendicular to line segment CB.
...

**Summary:**
The figure depicts a triangle ABC and triangle CBD, which is similar to triangle ABC. Additionally, triangle ACB is also present in the figure. The lengths of the segments CB, CD, and BD are 5, 3, and 4 units respectively. Point D lies on the line segment AB, and perpendicularity is indicated between the line segments AC and CB, as well as between the line segments CD and AD.

---

*Phase-2 Generation*

**QA pairs:**
1. Question: Does the diagram include a triangle formed by vertices A, B, and D?
Answer: **No**, the diagram does not include a triangle formed by vertices A, B, and D. The triangles present are ABC and CBD, and the point D lies on the line segment AB.
2. Question: Does the diagram contain a triangle with vertices A, C, and B?
Answer: **Yes**, the diagram contains triangle ACB, which is the same as triangle ABC.
3. Question: Is point D the lies on line BC?
Answer: **No**, point D is not on lines BC. The point D lies on the line segment AB.
4. Question: Is line segment AC perpendicular to line segment CB?
Answer: **Yes**, line segment AC is perpendicular to line segment CB.

---

Table 17: Geometric image description and factual QA pairs for understanding basic elements. The sections  in blue  display the responses generated by ChatGPT.

---

**Variable Substitution and Equation Solving**

**Question:**
Given a railway crossing barrier with two arms, where the shorter arm has a length of 'a' units (a = 1.0) and the longer arm has a length of 'b' units (b = 16.0), determine the elevation 'y' units of the longer arm's end when the shorter arm descends by 'd' units (d = 0.5). Express your answer as a function of 'a', 'b', and 'd'.

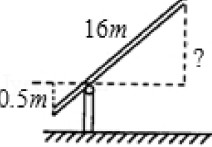

**Answer:**
To find the elevation 'y' of the long arm's end, we apply the property of similar triangles and establish the equation $\frac{d}{y} = \frac{a}{b}$. Substituting the given values, we get $\frac{0.5}{y} = \frac{1.0}{16.0}$. Solving for 'y' leads to $y = \frac{b \times d}{a}$, which simplifies to $y = \frac{16.0 \times 0.5}{1.0}$. Therefore, 'y' equals 8.0 units, indicating that the long arm rises by 8.0 units when the short arm descends by 0.5 units.

---

**Question:**
In a railway crossing lever system, the lengths of the shorter and longer arms are 's' units (s = 1.0) and 'l' units (l = 16.0) respectively. If the shorter arm lowers by 'deltaS' units (deltaS = 0.5), what is the increase 'deltaL' in the length of the longer arm?

**Answer:**
The increase 'deltaL' in the longer arm can be found using the proportionality relationship of similar triangles. The equation can be expressed as: $\frac{deltaS}{deltaL} = \frac{s}{l}$. Inserting the given values, the equation becomes $\frac{0.5}{deltaL} = \frac{1.0}{16.0}$. Solving for 'deltaL' yields $deltaL = \frac{l \times deltaS}{s}$, which simplifies to $deltaL = \frac{16.0 \times 0.5}{1.0}$. Thus, deltaL is 8.0 units.

---

**Question:**
At a railroad crossing, the short arm of the barrier is 1.0 unit long and the long arm is 16.0 units long. If the short arm drops 0.5 units, by how many units will the long arm rise?
Represent the rise of the long arm as 'h'.

**Answer:**
The triangles formed by the shorter and longer arms as they descend are similar. The rise of the long arm 'h' can be calculated using the property of similar triangles, the equation is 0.5/h = 1/16. Solving for 'h' gives h = 16 * 0.5 / 1, which equals 8.0 units.

Table 18: Synthetic example via replacing values to unknown variables and solving the equation.

---

**Prompt for Generating Geometric Image Description**

As a specialist in geometric mathematics with expertise in multi-modal (vision-language) processing, your skill involves reimagining diagrams based on the information provided in questions and answers. I will provide you with a question-answer pair related to a geometric diagram, and your task will be to mentally reconstruct the corresponding diagram and provide a description without involving calculations or mentioning specific numerical values.
To get started, let's consider three examples:

QA pair:
Question: [Sample Question]
Answer: [Sample Answer]
Description: [Sample Description]
...

I need a correct and comprehensive description of the diagram, avoiding any specific numerical data such as exact measurements or angles. Clearly identify the points within the shapes (e.g., on the circumference of the circle).
You should not mention details specific to this given question.

Table 19: Prompt for Generating Alignment Data.

---

**Prompt for Generating Factual QA Pairs for Basic Elements: Phase-1 Generation**

You are a helpful expert in geometric mathematics. I will first provide you with few-shots that each contain a question and the associated correct answer. Then, you should answer a new question.

Example 1:
Question:
logic_form:
[Sample Logic Form]

Above are logical forms for a geometric figure. You should describe the information in this figure.

Answer:
The geometric figure has the following information:
Info 1: [Sample Info 1]
Info 2: [Sample Info 2]
...

Summary:
[Sample Summary]

Example 2:
...

You must response in the following template:

The geometric figure have the following information:
Info 1: <INFO>
Info 2: <INFO>
Info 3: <INFO>
...
Summary:
<SUMMARY>

---

Table 20: Prompt for generating factual QA pairs for basic elements in Phase-1 Generation.

---

**Prompt for Generating Contrastive QA Pairs for Basic Elements: Phase-2 Generation**

As an expert in geometric mathematics, I will provide you with a detailed description of a geometric diagram. Your task will be to formulate contrastive questions and answers related to this diagram, such as 1) whether the geometry has triangle elements? 2) whether the description of the relationship between point and line is correct?

The questions should be directly related to the described diagram and should be answerable based on the provided geometric information without requiring numerical calculations.

logic_forms:
[Sample Logic Form]

info_items:
The geometric figure has the following information:
Info 1: [Sample Info 1]
Info 2: [Sample Info 2]
...

Summary:
[Sample Summary]

QA pairs:
pair_1:
Question: [Sample Question]
Answer: [Sample Answer]

pair_2:
...

Now I will give you the geometric information. You should give questions and answers related to this diagram. The questions should pertain to the specifics of the diagram, and they must be answerable using the given geometric details.

You should answer using the following template:

QA pairs:
pair_1:
Question: <QUESTION>
Answer:<ANSWER>

pair_2:
Question: <QUESTION>
Answer:<ANSWER>

pair_3:
Question: <QUESTION>
Answer:<ANSWER>
...

---

Table 21: Prompt for generating factual QA pairs for basic elements in Phase-2 Generation.

**Prompt for Equation Solving (ES)**

As a proficient in geometric mathematics, please rephrase both the question and the answer utilizing your advanced knowledge in the field. You should substitute variables with new ones such as x, y, z, or any other suitable replacements you can envision to solve the question. Furthermore, in certain steps, you should attain the solution by solving an equation involving these variables.

Here are the examples:

Current question-answer pairs:
Question: [The Original Question]
Answer: [The Original Answer]

New question-answer pairs:
Rephrased question: [New Question]
Rephrased answer: [New Answer]
...

Please rephrase both the question and the answer utilizing your advanced knowledge in the field following the above example. You should end the answer with "Result:[RESULT]".

Table 22: Prompt for equation solving stategy.

**Prompt for Value Scaling (VS)**

As a specialist in increasing numeric values, your role is to adjust the numbers in both the question and the answer by proportionally enlarging them, for instance, by multiples like 2 times.
Example:

QA pairs:
Question: [The Original Question]
Answer: [The Original Answer]

New QA pairs:
Rewritten question:[New Question]
Rewritten answer: [New Answer]
...

You should ensure that the solution is accurate. Conclude your response with: 'Therefore, the answer is option <CHOICE>'. Focus on scaling up values, rather than performing calculations. You should end the answer with "Result:[RESULT]".

You should answer using the following template:
New QA pairs:
Rewritten question: <QUESTION>
Rewritten answer: <ANSWER>

Table 23: Prompt for value scaling strategy.

---

**Prompt for Re-Formulating Condition as Unknown (RCU)**

As a geometric mathematics expert, please rephrase both the question and the answer based on a given question-answer pair. I will replace one known variable in the question with 'u' to create a new question, your task is to solve it using other information in the previous question-answer pair.

I will also provide you with the full information. You should solve the new question and promise the solution is correct.

Example:
QA pairs:
Previous question: [The Original Question]
Previous answer: [The Original Answer]

Full information:
[The Original Full Information]

New QA pairs:
New question: [New Question]
New answer: [New Answer]

You should finish the <ANSWER> in the new answer. You should end the answer with "Result:[RESULT]".

You should respond using the following template:
New answer: <ANSWER>

---

Table 24: Prompt for Re-Formulating Condition as Unknown (RCU) strategy.

---

**Prompt for Sentence Paraphrase (SP)**

You are an AI assistant to help me rephrase questions. You should rephrase my questions to be different with complete information. Here are some examples:

Question: [The Original Question]
Rephrase the above question: [Rephrased Question]
...

You should help me rephrase a new question following the given examples. For each question, you should rephrase to 10 new questions in the following format:

Question: <RAW_QUESTION>
Rephrase the above question:
Q1.<REPHRASED_QUESTION>
Q2.<REPHRASED_QUESTION>
...

---

Table 25: Prompt for sentence paraphrase strategy.

