# OpenReview forum: "G-LLaVA: Solving Geometric Problem with Multi-Modal Large Language Model"
_ICLR.cc/2025/Conference — ICLR 2025 Poster_

### Official Review · Reviewer_UDES · 2024-10-28

**Soundness:** 2
**Presentation:** 3
**Contribution:** 2
**Rating:** 6
**Confidence:** 4

**Summary:**

The paper is a pioneering work about Geometry Problem Solving using MLLM. The contribution is to provide a dataset and also a baseline using LLaVA.

**Strengths:**

It's a pioneering work on MLLM's mathematical reasoning especially focusing on geometry problem. Compared with its earlier version, the authors added more experiments on recent datasets like MathVerse.

**Weaknesses:**

The work was a milestone in developing mathematical skills of MLLM and the authors tried to append results on a latest dataset. However, the research in the area moves fast in both dataset construction and methods.
Despite the updates, the paper is still flawed in
1. Its targeted problem, i.e., geometry problem solving (GPS), is limited nowadays. The mathematical reasoning of MLLM evolves fast with the corresponding dataset. Now, it also covers the function plots in MathVerse and chart in MultiMath.
2. The baseline needs a fair comparison. Even though the problem is confined to GPS and methods in comparison is confined to open-source methods, the methods like MathLLaVA, MultiMath and Math-PUMA are encouraged to be compared with.

**Questions:**

Please refer to the Weaknesses for more details. All in all, the authors are encouraged to append the discussion or experiments about recent progress in both dataset and methods.

---

### Official Review · Reviewer_Nr4G · 2024-10-31

**Soundness:** 3
**Presentation:** 3
**Contribution:** 3
**Rating:** 5
**Confidence:** 5

**Summary:**

This paper investigates the capabilities of existing MLLMs in geometric problem-solving tasks and introduces the Geo170K dataset, which contains over 170K image-caption and QA pairs. It also proposes training the G-Llava using the proposed dataset and validates its performance on the GeoQA dataset, demonstrating its effectiveness.

**Strengths:**

1. This work explores an interesting topic, geometric problem solving, and currently, most MLLMs perform poorly on this task.
2. The expression and organization of this manuscript make it easy for readers to understand.
3. The dataset proposed in this manuscript is relatively large and includes image-caption and question-answer pairs data.

**Weaknesses:**

1. Novelty and Technical Contribution: The proposed geometric dataset construction approach relies on the existing logical annotation data and uses ChatGPT for automatic labeling. Besides, the overall model structure and training strategy mainly apply the existing techniques, lacking sufficient innovation. Please explain the differences in model structure or training strategy during the rebuttal phase and whether the authors made customized designs on a geometric domain. No specific considerations for the geometric domain were found.
2. Data Quality Issues: Since annotations are generated automatically by ChatGPT, is there any hallucination in the dataset? Is there any quality inspection conducted to support the claim of a "high-quality dataset"?
3. Insufficient Evaluation: The paper lacks quantitative comparisons on geometry-task-related benchmarks like UniGeo, Geometry3K, and PGPS9K, commonly used in prior studies [1-4].

[1] Chen J, Li T, Qin J, et al. Unigeo: Unifying geometry logical reasoning via reformulating mathematical expression[J]. arXiv preprint arXiv:2212.02746, 2022.
[2] Liang Z, Yang T, Zhang J, et al. Unimath: A foundational and multimodal mathematical reasoner[C]//Proceedings of the 2023 Conference on Empirical Methods in Natural Language Processing. 2023: 7126-7133.
[3] Zhang M L, Yin F, Liu C L. A multi-modal neural geometric solver with textual clauses parsed from diagram[J]. arXiv preprint arXiv:2302.11097, 2023.
[4] Li Z Z, Zhang M L, Yin F, et al. LANS: A layout-aware neural solver for plane geometry problem[J]. arXiv preprint arXiv:2311.16476, 2023.

**Questions:**

Please check the previous sections.

---

> ### Comment · Reviewer_tzwZ · 2024-12-03
>
> Thank you for your response. Constructing a dataset is indeed important, but there should be quantitative metrics for data quality validation. Additionally, evaluations should be conducted on multiple mainstream MLLMs. For models with open-source SFT, fine-tuning is needed; for others, direct testing is sufficient. These steps are crucial for a comprehensive experiment. We maintain the original score.

---

### Official Review · Reviewer_tzwZ · 2024-11-01

**Soundness:** 3
**Presentation:** 3
**Contribution:** 2
**Rating:** 5
**Confidence:** 4

**Summary:**

This paper presents G-LLaVA, an MLLM focused on geometric problems. The authors propose that the difficulty in geometric problems lies in logical structure compactness. Subsequently, they introduce a new dataset, Geo170K, with geometric cross-modal alignment data and geometric instruction data to enhance the geometric perception ability of MLLM.

**Strengths:**

1. Understanding geometric problems is important for MLLMs.
2. Presents a dataset with geometric alignment and instruction data.

**Weaknesses:**

1. The novelty is limited, as it primarily fine-tunes an existing MLLM.
2. The models compared in Table 14 need to be updated; stronger baselines, such as QwenVL2, InternVL2 should be considered.
3. Since the authors claim that G-LLaVA enhances geometric understanding from image input, it would be interesting to see the G-LLaVA's  performance improvement when only input $Q$ is provided, in Table 8, 9 and 14.

**Questions:**

In Table 11, why does just including SP increase accuracy by 8%? This type of data augmentation doesn't enhance the model's understanding of geometry. Does this imply that the performance gains are due more to an improved understanding of the problem description rather than geometry?

---

### Official Review · Reviewer_1KS1 · 2024-11-02

**Soundness:** 3
**Presentation:** 4
**Contribution:** 3
**Rating:** 6
**Confidence:** 3

**Summary:**

Current Multi-Modal Language Models (MLLMs) struggle to solve geometry problems for lack of comprehension of geometry information. This paper generates a large-scale geometric problem dataset named Geo170K by GPT3.5, which contains more than 170K geometric image-caption and QA pairs. The data is made up of two components: the alignment data and instruction data. The former is generated through logical forms in Geometry3K while the latter augments QA pairs in GeoQA+. Besides, a model based on LLaVa is proposed to be trained on Geo170K. Numerical results show that G-LLaVA outperforms existing traditional methods and MLLMs like GPT4-V on the geometry problem solving (GPS) tasks.

**Strengths:**

- This paper curates a large-scale geometry dataset (28 times larger than the biggest existing geometry dataset) containing both detailed description of geometric images and reasoning paths, which may help MLLMs improve their ability in geometric comprehension.
- Data augmentation techniques like value scaling, equation solving, Re-Formulating conditions as unknown, and sentence paraphrase make it convenient to enlarge the size of dataset and enhance robustness.
- Extensive experiments validate G-LLaVA’s superiority in geometry problem solving. Experiments on MathVerse demonstrate its generalization ability.

**Weaknesses:**

- The experiments are all about choice questions. Some works also involve proving[1] and completion[2] questions, which can be more difficult. Experiments on such questions are neglected.
- In many experiments, the strongest existing model GPT4-V is not included, making the result less convincing.
- The proposed G-LLaVA selects LLAMA-2 as LLM and a pretrained ViT as the vision encoder. A comparative analysis or experiment on other architecture or pretrained model may help justify this choice.

[1] Chen J, Li T, Qin J, et al. Unigeo: Unifying geometry logical reasoning via reformulating mathematical expression[J]. arXiv preprint arXiv:2212.02746, 2022.

[2] Li Z Z, Zhang M L, Yin F, et al. LANS: A layout-aware neural solver for plane geometry problem[J]. arXiv preprint arXiv:2311.16476, 2023.

**Questions:**

1. Will the RCU process generate a question with multiple answers, since sometime the cause and effect are not necessarily sufficient and necessary conditions? And unfortunately, the data correctness verification cannot recognize such case.
2. In Table 10, the performance gain led by cross-modal alignment seems limited, and why the fixed LLM performs better than tunable LLM?

---

### Meta-Review · Area_Chair_cSWA · 2024-12-22

**Metareview:**

This paper introduces G-LLaVA, a Multimodal Large Language Model fine-tuned with Geo170K, a dataset of 170,000 geometric image-caption and QA pairs. It is recognized as a pioneering effort in addressing geometric problem-solving in MLLMs, with an innovative data generation approach that effectively utilizes geometric logic, demonstrating scalability and substantial performance improvements.

While concerns were raised about methodological novelty and data quality assurance, the authors provided thorough explanations and evidence to address these issues. They emphasized that the primary contribution lies in the customized data generation tailored to the geometric domain, enabling effective training without altering the model architecture. The adoption of their dataset by leading institutions further highlights its value.

This paper makes a significant contribution to advancing MLLMs' geometric reasoning capabilities through an innovative, data-centric approach. It addresses a critical gap in the field and offers valuable resources for future research.

**Additional Comments On Reviewer Discussion:**

During the rebuttal period, the authors addressed key concerns raised by the reviewers. Reviewer 1KS1 and Reviewer Nr4G questioned the novelty and dataset quality. Reviewer 1KS1 highlighted the lack of architectural innovation, to which the authors clarified that their primary contribution lies in an innovative data generation pipeline, enabling high performance without modifying the model architecture. Reviewer Nr4G raised concerns about dataset validation, and the authors detailed their correctness verification process, emphasizing the dataset’s quality through benchmark improvements and its adoption by leading institutions.

Reviewer UDES and Reviewer tzwZ focused on the evaluation scope. Reviewer UDES noted the absence of comparisons with newer baseline models, which the authors explained were released after their work was completed. They included comparisons with mainstream MLLMs, such as GPT-4V, to demonstrate their model's effectiveness. Reviewer tzwZ raised concerns about the narrow focus on multiple-choice problems, and the authors demonstrated the model’s generalizability with datasets covering a broader range of question formats.

Overall, the authors provided thorough responses and evidence supporting the significance of their contributions. While some concerns remained, the data generation approach and substantial performance improvements were recognized as significant advancements.

---

### Decision · Program_Chairs · 2025-01-22

Accept (Poster)